# A Scoping Review of Food Insecurity and Related Factors among Cancer Survivors

**DOI:** 10.3390/nu14132723

**Published:** 2022-06-29

**Authors:** Courtney A. Parks, Leah R. Carpenter, Kristen R. Sullivan, Whitney Clausen, Tony Gargano, Tracy L. Wiedt, Colleen Doyle, Kanako Kashima, Amy L. Yaroch

**Affiliations:** 1Gretchen Swanson Center for Nutrition, Omaha, NE 68154, USA; lcarpenter@centerfornutrition.org (L.R.C.); wclausen@centerfornutrition.org (W.C.); tgargano@centerfornutrition.org (T.G.); ayaroch@centerfornutrition.org (A.L.Y.); 2American Cancer Society, Prevention and Early Detection, Patient Support, Atlanta, GA 30329, USA; kristen.sullivan@cancer.org (K.R.S.); tracy.wiedt@cancer.org (T.L.W.); colleen@colleenmdoyle.com (C.D.); kanakokashima@gmail.com (K.K.)

**Keywords:** food insecurity, cancer survivors, patient navigators, psychosocial impact, caregiver

## Abstract

Despite growing awareness of the financial burden that a cancer diagnosis places on a household, there is limited understanding of the risk for food insecurity among this population. The current study reviewed literature focusing on the relationship between food insecurity, cancer, and related factors among cancer survivors and their caregivers. In total, 49 articles (across 45 studies) were reviewed and spanned topic areas: patient navigation/social worker role, caregiver role, psychosocial impacts, and food insecurity/financial toxicity. Patient navigation yielded positive impacts including perceptions of better quality of care and improved health related quality of life. Caregivers served multiple roles: managing medications, emotional support, and medical advocacy. Subsequently, caregivers experience financial burden with loss of employment and work productivity. Negative psychosocial impacts experienced by cancer survivors included: cognitive impairment, financial constraints, and lack of coping skills. Financial strain experienced by cancer survivors was reported to influence ratings of physical/mental health and symptom burden. These results highlight that fields of food insecurity, obesity, and cancer control have typically grappled with these issues in isolation and have not robustly studied these factors in conjunction. There is an urgent need for well-designed studies with appropriate methods to establish key determinants of food insecurity among cancer survivors with multidisciplinary collaborators.

## 1. Introduction

Despite overall declining incidence rates in men and stable rates in women, the number of cancer survivors continues to grow in the United States (US), underscoring the importance of addressing health related quality of life (HRQOL) and food insecurity in this population [1]. Food insecurity is defined as the lack of consistent access to nutritionally adequate and safe food acquired in socially acceptable ways [2]. The impact of food insecurity is essential for considering among cancer survivors that experience a financial burden coupled with potential immunosuppression and need for adequate nutrition [3,4]. However, there has been very little integration of research focusing on food insecurity among low-income cancer survivors and the relationship to related psychosocial outcomes.

Individuals with lower socioeconomic status carry a greater proportion of the burden of obesity and cancer incidence and mortality [5,6,7,8]. These disparities indicate inequalities in cancer screening, dietary patterns, physical activity, and other health behaviors [7]. Broadly across the literature, emphasis in research among low-income populations and cancer has typically been placed on best practices for cancer screening and treatment [9,10,11]. Existing reviews across a general population of cancer survivors with some emphasis on low-income populations have addressed factors related to employment [12,13,14], self-management and psychosocial interventions [15], and financially burdened family caregivers [16]. Consequently, there remains sparse evidence regarding the role of diet and food insecurity in cancer prevention and control but there is literature more specific to cancer survivors. The purpose of this scoping literature review is to examine the relationship between food insecurity, cancer, and related factors among cancer survivors and their caregivers to inform programmatic and policy efforts.

## 2. Materials and Methods

### 2.1. Data Sources

A search of three bibliographic databases was conducted in November of 2018 (Pubmed, EBSCO/CINAHL, and PsychINFO). The search was limited to studies in English published since 1980 that were specific to the US and other English-speaking high-income countries. Searches were conducted using medical subject headings and synonyms for food insecurity and cancer and English-speaking high-income countries. A final step was to conduct a hand search of reference lists for additional relevant titles and Google Scholar to ensure that all relevant literature was included.

### 2.2. Study Selection

All identified citations were uploaded to DistillerSR (tool that facilitates reviews with greater transparency and audit-ready results; Evidence Partners, Ottawa, ON, Canada). Processes that followed were completed in DistillerSR and include title-, abstract-, and full-text level screening, and data extraction. Exclusion criteria during title and abstract screening stages included manuscripts that were literature reviews/metanalyses, letters to the editor, commentaries, studies conducted in a non-English-speaking country, and titles that were conference proceedings. 

Specific sub-topics were retained that were most relevant for the current literature review based upon a lack of existing reviews, which included, food insecurity or financial impacts of cancer; the role of caregivers; patient navigation. During full-text data extraction, it was determined that certain sub-topics were outside of the scope of this review or have several existing reviews published [12,13,14,15,16,17,18]. The topics that were excluded at this stage included: the impact of poverty on cancer, disparities in cancer treatment, cancer screening among underserved populations, and work/employment factors (applied to 355 articles). An additional 120 articles were excluded at the full-text data extraction phase due to earlier stage exclusion criteria that were not detected, yielding a final 49 articles across 45 studies summarized in this review. Figure 1 shows the process for study selection and review and the number of papers that were excluded at each stage.

### 2.3. Data Extraction

Title and abstract screening were performed for each article by one of four authors (C.P., L.C., W.C., T.G.), after several rounds of consensus building, full-text screening on the remaining titles was performed by two independent reviewers (combination of: C.P., L.C., W.C., T.G.) to determine inclusion or exclusion from the literature review with conflicts resolved between these four authors. The full-text data extraction in DistillerSR including fields: country, purpose, measurement tools, location (geography, institutions, rural/urban), design, study population, summary of results, implications. 

## 3. Results

### 3.1. Study Selection

The initial search yielded 3128 references that were added to DistillerSR software from PubMed (*n* = 1908), EBSCO/CINAHL (*n* = 603), PsychINFO (*n* = 590), and Google Scholar (*n* = 27). An integrated duplication detection tool was used to identify redundant citations. Duplicates were removed (*n* = 549), leaving 2759 titles to screen for further exclusion criteria. Following an initial title screen, 1257 references advanced to the abstract screen. In the abstract screening phase, the full abstracts were reviewed for further detection of exclusion criteria, which resulted in 524 of the titles being retained. At the full-text data extraction stage, 355 articles were excluded because their topic areas were deemed outside of the scope of the current review (employment and return to work, *n* = 87; cancer screening and prevention, *n* = 184; cancer treatment disparities, *n* = 48; impact of poverty, *n* = 25; other, *n* = 11). 

### 3.2. Study Characteristics

The 49 articles across 45 studies selected for inclusion spanned three different main topic areas: patient navigation (PN) and social worker role (*n* = 12); caregiver role and impact (*n* = 9); psychosocial impacts of cancer (*n* = 16); food insecurity and financial toxicity (*n* = 12). These 45 studies spanned countries where the studies took place included: Australia (*n* = 2); Canada (*n* = 4); the US (*n* = 37); and two of the US-based studies also had component conducted in Australia.

#### 3.2.1. PN and Social Worker Role

In total, there were 12 papers in this topic area (Table 1), seven of which reported on the characteristics and outcomes from PN pilots/trials [19,20,21,22,23,24,25], while five reported descriptive findings to inform PN strategies [26,27,28,29,30]. From the studies that described outcomes, one reported that PN did not yield significant improvements in patient outcomes when compared to controls or standard of care [22]. However, other studies did report positive impacts of PN on various outcomes including: time to resolution following abnormal screening results [23], receiving better quality of care [24], receipt of treatment for depression and improved HRQOL [19,20,21], and a reduction in perceived distress [25]. In a randomized controlled trial (RCT) to test the effects of PN on adherence to treatment for depression among breast cancer survivors, the intervention group had more favorable depression scores and social and functional well-being at 12 months [20]. However, at 24 months, intervention and control groups received similar amounts of treatment and depression recurrence was similar between groups [19]. Finally, this RCT demonstrated that rates of unemployment, medical cost and wage concerns, and financial stress were stable through 6 months, followed by a pronounced drop at 12 months for the PN intervention group [21]. Patients reporting economic concerns had significantly poorer functional, emotional, and affective well-being. Qualitative results from this study described negative economic changes precipitated by cancer diagnosis (e.g., income decline, under employment, economic stress) [21].

The remaining PN studies did not include comparison groups and were descriptive in nature [28,29,30]. Retention strategies for a depression treatment program among survivors addressed barriers such as: provision of information, patient-provider relationship, and instrumental strategies (e.g., providing transportation) [30]. A study characterizing patients in a cancer nutrition rehabilitation program found that survivors with fewer psychosocial problems tended to be older (i.e., 63–94 years) and a common reason for referral to social workers was for assistance with emotional problems and coping skills related to their illness [29]. Similarly, increased age and minority race-ethnicity status were associated with higher satisfaction with cancer care among survivors receiving PN [28]. Formative research to inform PN strategies described the most common needs and services of cancer survivors, and how the network of agencies providing resources are connected [26,27]. The needs of cancer survivors revealed through focus groups included: (1) improved access to quality care; (2) emotional and practical concerns; (3) family concerns; (4) PN involvement across the continuum of care [26,27]. The composition and function of a network of service providers to help improve connectivity and referrals between agencies found that those providing informational services (e.g., health education) were more likely to refer patients and there was a need for more specialized services (e.g., prostheses, housing) [27]. 

#### 3.2.2. Caregiver Role and Impact

In total there were nine papers (across eight studies) that focused on the role that caregivers play in supporting cancer survivors and impacts experienced by households (Table 1) [5,31,32,33,34,35,36,37,38]. Seven of these papers described qualitative or quantitative results from parents of children with cancer [31,32,33,34,35,36,37]. Focus groups with parents revealed both mothers and fathers reporting multiple dimensions of caregiving for their sick child (e.g., managing medications, emotional management) and parents relayed that these activities can become a full-time job, leaving little time for other activities or employment [32,33,34]. In addition, medical advocacy was described as a necessity to overcome barriers in the health care system and to ensure their child was receiving the best care possible [33]. Quantitative survey results among parents from these studies demonstrated that financial burden and hardship was experienced by parents of child cancer survivors, especially by those with lower incomes [31,35]. This financial burden was associated with loss of employment and work productivity, even for those surveyed in countries with universal health care (i.e., Canada, Australia) [35,36,37]. In addition, parents reported that loss of work productivity was associated with anxiety, depression, and negative health outcomes for themselves [38]. Families reported coping with financial hardship by fundraising and reducing spending to offset the cost of treatment [35]. Parents of children that had cancer reported the challenges that their now young adult children faced as a result of their experience as cancer survivors [36]. Some of these challenges included difficulty with employment due to disability, lack of employer support, the need for assistance from family members for activities of daily living, and financial assistance for basic necessities (e.g., clothing, food) [36].

Caregivers supporting cancer survivors also reported experiencing emotional and financial burden [5,38]. Those with greater work productivity loss tended to have increased caregiving hours, be caring for a loved one with more advanced cancer, to be married, and to report greater anxiety, depression, and burden related to financial problems [38]. Some caregivers reported that their own health suffered because they could not find the time or resources to visit a doctor when they needed [38]. One study found that surprisingly, a clearly terminal (negative) prognosis facilitated clear priorities, unambiguous emotion management, and improved social bonds while a more ambiguous (positive) prognosis fostered role conflict and clashing feelings with ongoing guilt within spousal caregivers [5].

#### 3.2.3. Psychosocial Impacts

16 papers across 15 studies were found that described the psychosocial impacts experienced by cancer survivors (Table 1) [39,40,41,42,43,44,45,46,47,48,49,50,51,52,53,54]. Intervention studies addressed HRQOL, stress management, and information provision [39,48,49]. A culturally sensitive telephone counseling intervention with Latina cervical cancer survivors yielded improvements in physical well-being, HRQOL, social/family wellbeing, and positive emotional effects [39]. An online health consultation to breast cancer survivors found the program improved self-efficacy, and positive perceptions of the doctor–patient relationship [49]. However, the online counseling intervention did not significantly change information seeking or perceived social support [48]. Similarly, an intervention with African American breast cancer survivors found that psychological well-being and HRQOL did not differ between the intervention and control groups, and overall improvements were attributed to factors the study could not account for [48]. A comparative study assessing differences across disease stage of prostate cancer found that men with less education experienced greater improvement in their mental well-being than did men with more than a high school education [41].

Qualitative studies with cancer survivors explored the effects of chemotherapy on cognitive functioning (i.e., “chemo brain”) [40], the experience of financial burden in cancer treatment [44], psychosocial impacts of cancer on their families [50], reasons for dropout from a depression treatment program [54], and a mixed-methods study exploring needs of African American cancer survivors [53]. Despite the differences in the purpose of these qualitative studies, all of the findings help describe the psychosocial and practical challenges that cancer survivors face [40,41,50,53,54]. Some of the challenges faced by those who underwent chemotherapy included: cognitive impairment influencing ability to manage social and professional lives; financial constraints leading to missed, delayed, or limited treatment opportunities (including long-term survivorship); family stress and lack of coping skills to deal with the effects of cancer; the need to address an array of practical needs (e.g., transportation, financial), guidance on lifestyle information, post treatment plan, and social support [40,44,50,53,54].

Several of the descriptive quantitative studies focused on identifying HRQOL and psychosocial needs of cancer survivors [42,43,45,46,47,51]. When exploring the health promoting behaviors of low-income cancer survivors, it was found that various behaviors were employed (i.e., walking, maintaining a positive mental attitude, changing their diet) [51]. In addition, participants reported spirituality as important in maintaining a hopeful and positive outlook and a desire to learn more about feasible types of exercise, healthy eating, and stress management [51]. Low-income Latina cervical cancer survivors reported high levels of depression and that immigration-related stress was common (e.g., fear of deportation, navigating a foreign medical system) [43]. Cancer-related psychosocial resources, life stress, and optimism accounted for significant proportions of the variance in psychosocial outcomes [43]. A survey with uninsured men with prostate cancer found that men with spouses were more likely to have elected surgery and have better mental health, lower symptom distress, and higher spirituality than unpartnered participants [46]. This study also found that men with prostate cancer reported worse mental health than people with other chronic diseases and that spirituality and physical functioning were positively associated with mental health [45]. In terms of race and ethnicity, in a study among African American and Latino cancer survivors it was found that African American patients with unmet supportive care and health insurance needs were more likely to miss appointments compared to Latinos [42]. Amongst Latinos, legal health-related issues predicted missed appointments [42]. Latino men with prostate cancer tended to be less educated, more often in partnered relationships, and had more variable incomes compared with men of other ethnic/racial backgrounds [47]. A survey assessing the psychosocial needs among diverse underserved cancer survivors found that ethnicity was the sole predictor of needs, even after controlling for education, time since diagnosis, treatment status, marital status, and age [52]. The needs identified included informational (e.g., treatment); practical (e.g., finances, transportation); supportive (e.g., emotional/coping support); and spiritual [52].

#### 3.2.4. Food Insecurity and Financial Toxicity

Food insecurity and financial toxicity was described by 12 studies (Table 1) [55,56,57,58,59,60,61,62,63,64,65,66]. Survivors who reported more financial strain and burden as a result of cancer care costs were more likely to rate their physical and mental health poorer, have greater symptom burden, lower satisfaction with relationships, and lower HRQOL [55,58,59]. Those that were more likely to report experiencing material and psychological financial hardship (i.e., stress about their financial situation) were also more likely to be younger females, non-white, uninsured, treated more recently, have lower family income, and to have changed employment because of cancer [64]. Cancer survivors who reported cost-related medication nonadherence tended to be lower income, African American, and have non-employer-based medical insurance [61]. Cancer survivors that qualified for co-payment assistance reported engaging in lifestyle-altering coping strategies (spending less on leisure activities and basics like food and clothing, borrowing money, and spending savings) and care-altering coping strategies (not filling a prescription, taking less medication than prescribed) [62,65]. Participants with more education and shorter duration of chemotherapy reported using lifestyle-altering strategies more than their counterparts [62]. Two studies measured food insecurity among a sample of cancer survivors and found that these individuals had higher rates of food insecurity compared to the general population [57,63]. In addition, food insecure patients had significantly higher levels of nutritional risk (e.g., appetite, having only having liquids), depression, financial strain, lower HRQOL, and were more likely to not take prescribed medication because they reported not being able to afford it, compared to food secure patients [57,63]. Among a large cohort of cancer survivors, it was found that younger age, larger household size, and communicating with physicians about costs were associated with greater subjective financial burden [65]. 

Finally, descriptive studies reported findings that described the uptake of a novel emergency food system (i.e., food pantry within a hospital) [56] and the feasibility of an intervention to improve self-efficacy [60]. Results from the hospital-based food pantry pilot found that the mean number of return visits over a four-month period was 3.25 and that younger patients used the pantry less, immigrant patients used the pantry more (than US-born), and prostate cancer and later stage cancer patients used the pantry more [56]. Higher levels of education were related to higher levels of health-promoting behaviors such as reporting unusual signs or symptoms to their health professionals, questioning health professionals in order to understand instructions, and inspecting their bodies monthly for physical changes [60].

## 4. Discussion

Our scoping review revealed that food insecurity is an understudied challenge that is highly relevant for cancer survivors across the continuum. The American Cancer Society funded the establishment of a committee at the Institute of Medicine (IOM) to examine the range of medical and psychosocial issues faced by cancer survivors [67]. This consensus study suggests that with advances in cancer early detection and more effective treatment, long-term survivorship presents an opportunity to enhance the HRQOL across the continuum of cancer survivorship [67]. As the number of cancer survivors continue to grow and are living longer, it is important to consider the social determinants of health. Consideration of how social determinants of health (i.e., “upstream factors”) impact cancer survivors’ HRQOL will require multiple levels of analysis to understand the diverse pathways and mechanisms that link the social environment, healthcare delivery, and behavioral, psychological, and biological levels to develop more effective interventions [68]. Food insecurity may be considered a key social determinant of health [69] and is defined as the lack of consistent access to nutritionally adequate and safe food acquired in socially acceptable ways [2].

The financial impact on cancer survivors is vast and our review highlighted results from PN and social worker studies, caregiver role and impact, psychosocial impacts, and food insecurity and financial toxicity (Table 1). Only two studies in our review measured the construct of food insecurity and concluded that cancer survivors experience food insecurity at a higher rate than the general population and survivors experiencing food insecurity were also more likely to be at risk for nutritional deficiencies, depression, financial strain, and lower HRQOL [57,63]. In addition, multiple previous literature reviews concluded that there is significant employment loss for cancer survivors [12,13,14], which has logical implications for food insecurity, but has not been integrated into study methodologies. 

In addition to cancer survivors being at risk for food insecurity, there is some evidence to suggest that being food insecure may place an individual at increased risk for developing cancer. A report from the United States Department of Agriculture (USDA) using nationally representative data found that the prevalence of cancer increases as the severity of food insecurity increases [70]. A recent commentary examined the relationships between food insecurity and cancer and explored potential mechanisms and suggested several opportunities to address food insecurity among these individuals [71]. It is suggested that care providers can help identify food insecurity through screening and referral to relevant resources and intervention [71].

This review has several limitations that should be considered when interpreting the findings. A significant limitation was the lack of published research that measured food insecurity among cancer survivors, despite evidence that the financial impact of a cancer diagnosis on a household can be catastrophic. This limited focus on the construct of food insecurity led us to establish inclusion criteria that spanned related factors (i.e., PN and social worker studies, impact on caregivers, and psychosocial/HRQOL). Moreover, many of the studies had an overall high risk of bias due to several factors such as small sample sizes, confounding, missing data, cross-sectional design, and limited generalizability. Due to the heterogeneity of study designs, we did not use formal meta-analytic techniques.

## 5. Conclusions

Despite these limitations, this review suggests that there are multiple ways to improve the HRQOL of cancer survivors, and many potential areas of intervention to explore. Our scoping review highlights the state of the science on food insecurity and related factors among cancer survivors and summarizes determinants of financial burden and psychosocial outcomes. Future research may want to explore, develop, and test interventions that address food insecurity among cancer survivors. Some potential areas of intervention may include screening for food insecurity and referral to relevant resources, food pantries onsite at cancer clinics and other health care settings, incorporation of food insecurity efforts into patient navigation programs, and consideration of financial burden that cancer survivors face throughout the cancer survivorship continuum. Nutrition, public health, and cancer prevention and control fields have typically grappled with food insecurity, obesity, and cancer in isolation, and have not robustly studied these factors in conjunction. The number and complexity of the reported financial burdens that cancer survivors and their caregivers face suggest that there is an urgent need for well-designed studies with appropriate methods to establish key determinants of food insecurity.

## Figures and Tables

**Figure 1 nutrients-14-02723-f001:**
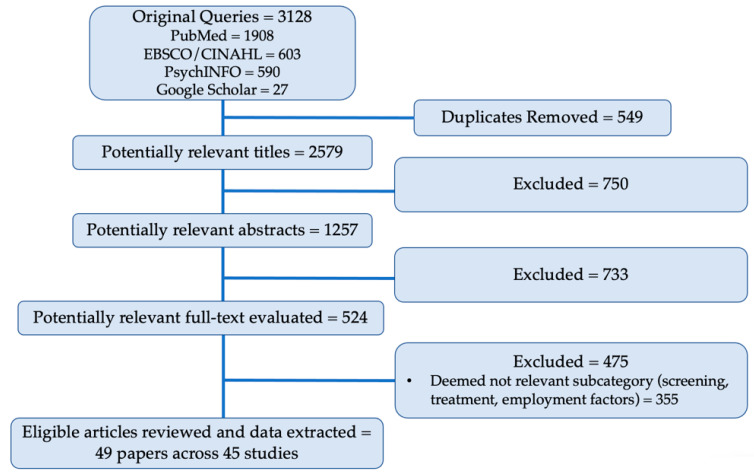
Flow diagram of inclusion for this review of food insecurity and related factors among cancer survivors. Pubmed, EBSCO/CINAHL, PsychINFO and Google Scholar are bibliographic databases.

**Table 1 nutrients-14-02723-t001:** Study Characteristics, Results Summary, and Implications.

Study/Year	Country	Purpose	Study Design/Participants	Summary of Results	Implications
**Patient navigation (PN) and social worker role**.					
Davis, 2009 [26]	US	Understand models of care facilitated by social workers.	Focus groups with cancer survivors in 3 Tennessee cities (*n* = 36).	PN programs need to address access to quality care, emotional, practical concerns. Participants reluctant to discuss emotional/practical distress with providers.	Oncology social workers have a unique opportunity to meet the needs of medically underserved survivors through PN.
Ell, 2007 [21]; Ell, 2008 [20]; Ell, 2009 [19]	US	Explore impact of economic stress on HRQOL of survivors and role of PN.	RCT involving surveys and follow-up interviews. Breast cancer patients at urban public safety-net medical center (*n* = 487 for RCT; *n* = 29 for interviews).	Unemployment/medical cost concerns, high at baseline, decline at 12 months. Lost wage worries increased baseline-6 months. Functional, emotional, physical, social-family well-being had positive linear improvement.	Managing depression among cancer survivors in a collaborative care system with PN shows promise. Future research can explore follow-up symptom monitoring and depression care in multi-site trials.
		Determine effectiveness of depression treatment.		12 months: 63% of intervention patients had ≥ 50% reduction in depressive symptoms; vs. 50% of control group.	
		Comparison of written resource navigation vs. written information plus PN.		24 months: 46% of intervention, 32% of control had ≥50% decrease in depression. Intervention patients had better social and functional well-being. Intervention patients more likely to receive treatment after 12 months.	
Harris, 2011 [27]	US	Describe referrals among organizations providing services to underserved cancer patients.	Survey with underserved cancer patients in St. Louis, MO, US (*n* = 33).	Those providing informational services were more likely to refer patients. Specialized services (e.g., prostheses) more likely to receive referrals. Few organizations provided housing services, smoking cessation, and were lacking in particular geographic areas.	Increased awareness building among provider organizations, broader geographic coverage, increased utilization of tobacco cessation, and financial assistance services are needed.
Jean-Pierre, 2016 [28]	US	Examine satisfaction with interpersonal relationship with PN.	Survey with cancer patients (*n* = 1345).	Increased age and minority race-ethnicity status associated with a higher satisfaction. Satisfaction with PN associated with satisfaction with cancer-related care.	There is a need to understanding role of interpersonal relationships and impact on care-related outcomes.
Martin, 2014 [22]	US	Determine impact of “I Can Cope” (ACS intervention).	Telephone survey among low-income cancer survivors (*n* = 140).	Intervention participants had lower information needs. Significant covariates with lower informational needs: higher self-efficacy, younger individuals, more educated, and higher income.	Supporting self-efficacy among cancer survivors may lead to lower information needs.
Raich, 2012 [23]	US	Evaluate the impact of lay PNs on time to resolution and follow-up testing.	Survey among medically underserved breast and prostate cancer survivors (*n* = 993).	PN positively impacts time to resolution of abnormal screening tests. Barriers experienced most by patients with low household income, low education attainment, high unemployment, high uninsured rate, and high comorbidity.	PN is an effective strategy for improving adherence to diagnostic evaluation and resolution, regardless of ethnicity, insurance status and education level.
Raj, 2012 [24]	US	Examine characteristics among women in a PN program.	Retrospective chart review of breast cancer patients (*n* = 168).	PN programs facilitated evidence-based quality care for vulnerable populations.	Need for improved processes and outcomes of PN in diverse underserved settings.
Townsend, 2010 [29]	Canada	Describe patients in a Cancer Nutrition Rehabilitation program.	Retrospective data from a cancer nutrition program (*n* = 75).	Lowest% of psychological problems were older survivors (63–94 years). Most patients (85.3%) were independent with activities of daily living. 55% needed assistance with basic needs (e.g., transportation, finances, groceries).	Supporting psychological needs for cancer survivors is important and may be more of a need among younger survivors.
Wells, 2015 [30]	US	Assess behavioral health intervention for retention among low-income minority cancer patients.	Interviews with behavioral health providers of a depression treatment program (*n* = 9).	Retention strategies clustered around five dropout barriers: (1) informational, (2) instrumental, (3) provider–patient therapeutic alliance, (4) clinic setting, and (5) depression treatment.	Further identification of simple, effective, feasible, and culturally sensitive means of retaining minorities in follow-up depression care is needed.
Wiggins, 2018 [25]	Australia	Evaluate social work service at a facility for cancer patients receiving treatment away from home.	Survey with cancer survivors utilizing a cancer council lodge while receiving treatment (*n* = 149).	Social work contact (*n* = 19) associated with reduction in distress, better able to manage challenges, and access services between arrival and departure compared with no contact (*n* = 56).	Type of support cancer survivors benefit from delivered by social workers has a wide range.
**Caregiver (CG) role and impact**					
Bona, 2016 [31]	US	Describe material hardship of families at a pediatric oncology center.	Surveys with families of children receiving chemo in MA, US (*n* = 99).	At baseline, 20% of families had low-income. At follow-up, work disruptions caused 25% of families to lose >40% household income and 29% to experience household material hardship.	Household material hardships are prevalent in newly diagnosed pediatric oncology families.
Clarke, 2005 [32]	Canada	Describe the health care activities of fathers of children with cancer.	Interviews with fathers of children with cancer (*n* = 18).	Home health care work of fathers included: monitoring and advocacy, collaboration with medical staff, scheduling, and administration/financial/emotional management.	Further research is needed on the work that fathers do when their children have cancer to inform specific tools for parents.
Clarke, 2006 [33]; Clarke, 2006 [34]	Canada	Describe the health care activities of mothers of children with cancer [33]. Describe the advocacy work these mothers engage in [34].	Focus groups with mothers of children with cancer (*n* = 49).	Caring for children was a full-time job and left no time for outside employment or other activities. Home health care activities included: managing medications/side effects, administration, and emotional work [33].	A paradigm shift is needed away from the psychological suffering to the strengths possessed by mothers.
				Medical advocacy for children carried out in response to perceived errors, understaffing, and peer advice. Many coped by educating themselves, extensive notes, and sharing their experiences with other parents [34].	
Dussel, 2011 [35]	US and Australia	Describe financial hardship, work disruptions, income loss, and coping of families.	Survey with bereaved parents of children lost to cancer (*n* = 230).	Financial hardship experienced by 24% (US) and 39% (Australia). Work disruptions led to reduction in income (60%). After accounting for income loss, 22% of families dropped below the poverty line. Fundraising and reduced spending were common coping strategies.	Existing health care, social, and work policies at three sites were not sufficient to prevent financial effects of a child’s death.
Howard, 2014 [36]	Canada	Describe parents of pediatric survivors perspectives of life challenges experienced by their now adult child.	Survey with parents of children that experienced a brain tumor in childhood (*n* = 46).	Participants had difficulty gaining/sustaining employment and independent living, some requiring continuous support. Support included help with grocery shopping and meal planning. Parents considered their children vulnerable and in need of protection.	Prospective longitudinal research is needed to explore factors such as: unemployment, financial challenges and legal difficulties, which appear to be poorly addressed by health and social programs.
Lau, 2014 [37]	US and Australia	Measure major life changes for parents at one year after child’s leukemia diagnosis.	Surveys (during the first 12 months of therapy) with parents of children with leukemia (*n* = 159).	Major life changes in first year of treatment is high: 13% divorced/separated, 27% relocated homes, 22% decided not to have more children, 51% declined job opportunities, 68% decreased work hours.	The steepest incidence of family burdens occurs at diagnosis. Social workers and others should help families anticipate these challenges and develop coping strategies.
Mazanec, 2011 [38]	US	Identify CGs differences in work productivity, CG burden, depression, anxiety, and social support.	Surveys with CGs (*n* = 69).	Work productivity loss for CGs associated with increased CG hours and cancer stage, marriage status, greater financial problems, disrupted schedule. 20% of CGs were unable to see their doctor when they needed. Work productivity loss related to anxiety, depression, financial burdens, disrupted schedule, health problems.	Health care providers are in a unique position to provide health promotion education to CGs, which may ultimately improve their health and reduce the economic impact of caregiving.
Olson, 2015 [5]	Australia	Understand variations in CGs emotional experiences.	Interviews with spousal CGs in Australia (*n* = 32).	A terminal (negative) prognosis facilitates clear priorities, unambiguous emotion management, and improved social bonds. An ambiguous (positive) prognosis fosters role conflict and ongoing guilt within spousal CGs.	To support CGs, it is imperative to consider characteristics of their experiences be examined to prepare health professionals to conduct psychological screening.
**Psychosocial (PS) impacts**					
Ashing-Giwa, 2008 [39]	US	Assess feasibility of implementing a culturally sensitive telephone intervention.	Surveys (pre-post) with Latina cervical cancer survivors. (Intervention *n* = 15, Control *n* = 8).	Increases in physical well-being and positive effects (e.g., outlook on life, hopefulness, energy level, family life, intimacy, spirituality, quality of care) found in intervention group. Social/family, emotional, and functional scores did not significantly improve in either group.	Intervention associated with changes in physical well-being and QOL.
Boykoff, 2009 [40]	US	Understand changes patients undergo to inform care.	Focus groups with African American cancer patients (*n* = 75).	Cognitive impairment influences interviewees’ ability to manage their social and professional lives. Coping strategies included use of tools such as notes and calendars, and having consistent behavioral patterns.	Greater knowledge of how “chemobrain” influences post-treatment HRQOL can inform strategies.
Brar, 2005 [41]	US	Investigate changes over time in general and disease-specific HRQOL.	Surveys with prostate cancer patients, <200% poverty level (*n* = 138).	Participants with advanced prostate cancer experienced more negative changes in health. Men with < high school education experienced greater improvement in mental well-being.	Findings from this study provide a unique view of HRQOL changes over time in the study population.
Costas-Muniz, 2016 [42]	US	Determine if unmet financial, logistic, and care needs predict adherence to cancer treatment.	Survey with low-income ethnic minority patients at a New York, NY, US cancer clinic (*n* = 1098).	≥4 unmet needs increased likelihood of reporting missing appointments. For African Americans, unmet supportive care and health insurance needs increased missed appointments. For Latinos, legal health-related issues were a predictor of missed appointments.	There is a need to understand the impact of practical and supportive unmet needs on adherence and development of interventions aiming to improve adherence.
D’Orazio, 2011 [43]	US	Explain the PS adjustment in order to describe and identify predictors of PS outcomes.	Survey with Latina cervical cancer patients from a California cancer clinic (*n* = 54).	Patients reported depression yet adequate amounts of social support. Cancer-related PS resources, life stress, and optimism accounted for PS outcomes. Common life stressors: fears of deportation, navigating a foreign medical system, not speaking English.	There is a need to develop explanatory models of adjustment for low-income Latina cervical cancer patients that include cancer-related and contextual predictors of PS well-being.
Darby, 2009 [44]	US	Explore the financial burden to inform culturally sensitivity.	Focus groups with African American survivors (*n* = 36).	Lack of insurance resulted in missed, delayed, or fewer treatment opportunities. Financial burden of cancer was not limited to the acute treatment phase.	Estimates regarding care costs should be interpreted with caution due to variations in measurement.
Gore, 2005 [45]; Gore, 2005 [46]	US	Evaluate the influence of partnership on HRQOL [45].	Survey with low-income, uninsured men participating in a state-funded program that provides free prostate cancer treatment (*n* = 291).	Partnered patients vs. unpartnered patients: Hispanic ethnicity (58% vs. 34%); More likely to have elected surgical therapy (49% vs. 34%); Better mental health, higher spirituality, lower symptom distress [45].	Single participants may represent an isolated cohort of men with prostate cancer. Coping and social support mechanisms to encourage the beneficial aspects of partnership and to overcome the detrimental effects of being single need to be addressed.
		Evaluate mental health outcomes in low income men with prostate cancer [46].		Prostate cancer patients report worse mental health. Hispanic ethnicity, urinary/bowel bother negatively associated with mental health. Spirituality and physical functioning positively associated with mental health [46].	
Krupski, 2005 [47]	US	Describe and compare HRQOL among men with localized prostate cancer.	Survey with low-income, prostate cancer patients (*n* = 208).	Hispanic men with prostate cancer were less educated, more often in significant relationships, had variable income, and had worse sexual and physical function compared to other ethnicities. African-American and Hispanic men were more spiritual than Caucasian men.	Attention to demographic variations in HRQOL may improve outcomes for low- survivors across ethnicities with specialized counseling and referrals to social support systems.
Lechner, 2014 [48]	US	Examine participation in a cognitive-behavioral stress management program.	Surveys/interviews with underserved breast cancer patients (*n* = 487; *n* = 29 for interviews).	Participants in both conditions showed improvement on psychological well-being, HRQOL, intrusive thoughts, depressive symptoms, and stress levels.	Lack of differences between the programs may be due to the natural course of PS improvement, actual improvement from the intervention, or a result of nonspecific factors.
Lu, 2011 [49]	US	Examine how PS variables predicted use of an online health consultation service and how use affected those same variables.	Pre-post examination of consultation services among low-income breast cancer patients below 250% poverty level (*n* = 231).	Online health consultation positively associated with three variables: health self-efficacy, participation in health care, and doctor–patient relationship. No significant relationships with information seeking and perceived social support variables.	Online health consultation complements other resources and increases confidence to participate in health care.
Marshall, 2011 [50]	US	Identify specific PS intervention needs of co-survivors.	Interviews with co-survivors on PS impacts (*n* = 16).	Themes: family stress, coping, need for financial help, and reliance on faith. Tailoring intervention to family needs and delivering it in accessible ways.	Outreach and engagement with various populations impacted by cancer, including co-survivors is important.
Meraviglia, 2011 [51]	US	Explore health-promoting behaviors of low-income cancer survivors.	Surveys with cancer patients from an urban cancer clinic (*n* = 51).	Health-promoting behaviors: walking, positive mental attitude, dietary changes, spiritual growth through prayer. Participants interested in learning about effective exercise, diet, and stress management.	Low-income cancer survivors engage in various health-promoting behaviors and want to learn strategies to use after treatment.
Moadel, 2007 [52]	US	Describe the development of a PS needs survey and patterns/predictors of need.	Survey with ethnically diverse underserved cancer patient population in Bronx, NY, US (*n* = 248).	Racial-ethnic minority cancer survivors have greater need for various PS supports: Informational; Practical (e.g., finances, transportation); Supportive (e.g., emotional/coping support); Spiritual (e.g., finding meaning/hope and spiritual resources).	There is a need to determine what interventions are most effective to address the informational, emotional, practical, and spiritual needs of these patients.
Mosavel, 2011 [53]	US	Identify the needs of low-income, African American cancer survivors in an urban setting.	Interviews/surveys with cancer survivors (*n* = 12), CGs (*n* = 10), professionals (*n* = 10); town halls (*n* = 80).	Participants identified practical needs (e.g., transportation, financial), lifestyle information, post treatment plan, and social support. The ideal resource would be located within the survivor’s neighborhood and provide medical support and recreational services.	Accrual of minorities in clinical cancer trials, attitudes/beliefs about cancer, and participation in research are issues that may be addressed by cancer resource centers within minority communities.
Wells, 2013 [54]	US	Explore patient perspectives about depression treatment.	Interviews with Latina cancer survivors in a depression treatment program (*n* = 30).	Treatment barriers: (a) barriers to treatment; (b) disease features; (c) treatment regimens; (d) provider–patient relationship; and (e) clinical setting. Completers more motivated and satisfied with treatment.	Need for educational approaches to address negative perception of antidepressants. Intensive case management useful.
**Food insecurity (FI) and financial toxicity**					
Fenn, 2014 [55]	US	Examine association between financial problems and reported HRQOL in a population-based sample of cancer patients.	Survey with secondary analysis of cancer survivors using NHIS data oversampling in minority population (*n* = 2108).	Degree to which cancer caused financial issues was the strongest predictor of HRQOL. Patients who reported that cancer caused “a lot” of financial problems were four times less likely to rate HRQOL as at least “good”. Lack of insurance is associated with the degree of cancer-related financial problems.	There is a need to give more attention to the economic burden of cancer and the impact on a patient’s overall well-being.
Gany, 2015 [56]	US	Examine predictors of us of a novel emergency food system at 5 clinics.	Survey with patients who visited hospital pantries in New York, NY, US (*n* = 351).	Younger patients used pantry less. Immigrant (non US-born), prostate cancer, and Stage IV cancer patients used the pantry more.	Cancer patients most at risk (e.g., immigrants, later stage cancers) need to be considered in the development of interventions to address FI.
Gany, 2015 [57]	US	Determine the relationship between FI and HRQOL.	Survey with ethnic minority cancer patients (*n* = 1390).	41.8% food secure, 41.1% with low food security, 17.1% very low food security. HRQOL decreased with food security level. Inverse relationship: physical, functional, social, emotional well-being with FI.	Minority cancer survivors at higher risk for FI and suffer lower HRQOL. Services to support food security among these survivors are needed.
Kale, 2016 [58]	US	Determine prevalence and assess predictors of cancer-related financial burden and HRQOL.	Survey with a national panel involving multiple agencies surveying cancer survivors. (*n* = 1380).	28.7% reported financial burden. Physical/mental HRQOL were lower for those with financial burden than those without. Survivors with financial burden had lower HRQOL, increased depression, and increased worry about cancer recurrence.	Future research should assess the role of value-based reimbursement, clinical practice guidelines, and physician-patient communication regarding reducing the cost of cancer care to help improve HRQOL.
Lathan, 2016 [59]	US	Measure association between financial strain, symptom burden, and HRQOL.	Survey with patients 5 Veterans Health Associations (*n* = 208).	Patients with lung (40%) and colorectal cancer (33%) reported limited financial reserves (≤2 months). Dose-response relationship was present across all measures of well-being with decreasing financial reserves.	Evaluation of financial strain could be performed by social workers, nurses, or physicians.
Meraviglia, 2015 [60]	US	Determine the feasibility of a health-promoting intervention.	Survey with low-income cancer survivors (*n* = 51).	>50% engaged in health-promoting behaviors (e.g., unusual symptoms, questioning health professionals, inspecting bodies for physical changes). Greater education related to health-promoting behaviors.	There is a need to understand the use of health-promoting behaviors and feasibility of interventions after treatment.
Nekhlyud, 2011 [61]	US	Compare cost-related medication nonadherence.	Survey with cancer survivors >65 (*n* = 9818).	Survivors who reported cost-related medication nonadherence tended to have lower income, be African-American, and have non-employer-based medication insurance.	Elderly Medicare cancer survivors may not face a greater perceived burden of medication costs than their peers.
Nipp, 2016 [62]	US	Describe patients at highest risk for using strategies to cope with treatment-related costs.	Survey with cancer survivors utilizing financial assistance from a non-profit organization (*n* = 174).	Younger patients more likely to use coping strategies. Strategies: spending less on leisure activities/basics, borrowing money, spending savings, not filling a prescription, and taking less medication than prescribed. Higher more education and shorter duration of chemotherapy used lifestyle-altering strategies more.	Qualitative assessments may help to better understand cancer survivors’ unique perspectives about financial burden.
Simmons, 2006 [63]	US	Examine the construct and correlates of FI in a sample of cancer patients.	Survey with patients at a university cancer clinic in Kentucky (*n* = 115).	FI rates higher than the general pop (25%). Patients with FI had greater nutritional risk, depression, financial strain, and lower HRQOL. 55% of FI patients did not take a prescribed medication because they could not afford it, versus 12.8% of food secure patients.	Understanding factors (including food insecurity) that may be associated with patient noncompliance is an important element of oncology care.
Yabroff, 2016 [64]	US	Estimate the prevalence of financial hardship associated with cancer.	Survey with cancer survivor’s data from the Medical Expenditure Panel Survey (*n* = 565).	Material psychological financial hardship greater among 18–64 than >65 years of age. Younger, female, nonwhite, treated recently, and changed employment because of cancer more likely to report financial hardship.	Further exploration on the financial hardship associated with cancer treatment as the health insurance landscape changes is needed.
Zafar, 2013 [65]	US	Describe experiences of patients using copayment assistance and the impact on well-being and treatment.	Surveys (pre and post) with patients that utilized a national copayment assistance foundation (*n* = 245).	Coping strategies: copayment assistance, reduced spending on food/clothing/leisure, used savings, partially filled prescriptions, avoided filling prescriptions. Greater financial burden associated with: younger age, larger household size, applying for copayment assistance, and communicating with physicians about costs.	Health insurance does not eliminate financial distress or health disparities among cancer patients. Financial distress as a result of disease or treatment decisions might be considered.
Zafar, 2014 [66]	US	Describe financial burden, disease status, HRQOL, comorbidities, and quality of care.	Survey from a large cohort of cancer survivors (*n* = 1000).	48% reported difficulties living on their household income. Financial burden associated with lower household income, younger age, and poorer HRQOL. Better HRQOL was associated with fewer perceptions of poorer quality of care.	Financial burden is prevalent among survivors and associated with HRQOL. Need for interventions to improve patient education and engagement with regard to financial burden.

HRQOL: health related quality of life; RCT: randomized controlled trial; ACS: American Cancer Society; NHIS: National Health Interview Survey.

## Data Availability

Not applicable.

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
