# Peer review of "A Scoping Review of Food Insecurity and Related Factors among Cancer Survivors"

_nutrients, 2022, doi:10.3390/nu14132723_

Round 1
Reviewer 1 Report
The review article entitled “A Scoping Review of Food Insecurity and Related Factors among Cancer Survivors” is well written and the results are well justified with proper citation of relevant references. This report may be of great interest to readers of the journal ‘Nutrients’. The manuscript should have after some minor corrections:
Comments:
- Will there be any differences on the ‘psychosocial impact’ of individuals living in a joint family or in a nuclear family? Can the data presented here be translated to justify this situation?
- Authors advised some limitations of the current study, household size can be discussed in relation to the financial and psychosocial impact.
- Page 10, line 225: ‘and negative affect…..’ should be written as ‘and negative effect…’?
Author Response
The review article entitled “A Scoping Review of Food Insecurity and Related Factors among Cancer Survivors” is well written and the results are well justified with proper citation of relevant references. This report may be of great interest to readers of the journal ‘Nutrients’. The manuscript should have after some minor corrections:
Comments:
- Will there be any differences on the ‘psychosocial impact’ of individuals living in a joint family or in a nuclear family? Can the data presented here be translated to justify this situation?
- Thank you for your review and comments. In looking through the articles included under "psychosocial impact" there were themes around social support and the role that caregivers play. However, there were not any specific findings on family structure (i.e., nuclear vs other family styles) that could be gleaned from these studies.
- Authors advised some limitations of the current study, household size can be discussed in relation to the financial and psychosocial impact.
- Household size was a factor in one large cohort study (contributing to greater perceived financial burden). This was described in line 267.
- Page 10, line 225: ‘and negative affect…..’ should be written as ‘and negative effect…’?
- This was intended as 'negative affect' (poor mood, etc.) but is very similar to the experience of depression, so this was deleted to avoid confusing the reader.
Reviewer 2 Report
In this scoping review, the authors aimed to identify of food insecurity and related factors among cancer survivors. The topic is interesting, and the study is well written; the Discussion is well organized, and the methodology is correct.
Author Response
In this scoping review, the authors aimed to identify of food insecurity and related factors among cancer survivors. The topic is interesting, and the study is well written; the Discussion is well organized, and the methodology is correct.
- Thank you for your view and comment. A few minor updates to the manuscript have been made.